# Surgical Management of Adrenocortical Carcinoma: Current Highlights

**DOI:** 10.3390/biomedicines9080909

**Published:** 2021-07-28

**Authors:** Giuseppe Cavallaro, Mariarita Tarallo, Ambra Chiappini, Daniele Crocetti, Andrea Polistena, Luigi Petramala, Simone Sibio, Giorgio De Toma, Enrico Fiori, Claudio Letizia

**Affiliations:** 1Department of Surgery “Pietro Valdoni”, Sapienza University, 00161 Rome, Italy; giuseppe.cavallaro@uniroma1.it (G.C.); ambra.chiappini@uniroma1.it (A.C.); daniele.crocetti@uniroma1.it (D.C.); andrea.polistena@uniroma1.it (A.P.); simone.sibio@uniroma1.it (S.S.); giorgio.detoma@uniroma1.it (G.D.T.); enrico.fiori@uniroma1.it (E.F.); 2Department of Translational and Precision Medicine, Sapienza University, 00161 Rome, Italy; luigi.petramala@uniroma1.it (L.P.); claudio.letizia@uniroma1.it (C.L.)

**Keywords:** adrenocortical carcinoma, current methods of treatment, laparoscopic adrenalectomy, robotic adrenalectomy

## Abstract

Introduction: Adrenocortical carcinoma (ACC) is a rare tumor, often discovered at an advanced stage and associated with poor prognosis. Treatment is guided by staging according to the European Network for the Study of Adrenal Tumors (ENSAT) classification. Surgery is the treatment of choice for ACC. The aim of this review is to provide a complete overview on surgical approaches and management of adrenocortical carcinoma. Methods: This comprehensive review has been carried out according to the PRISMA statement. The literature sources were the databases PubMed, Scopus and Cochrane Library. The search thread was: ((surgery) OR (adrenalectomy)) AND (adrenocortical carcinoma). Results: Among all studies identified, 17 were selected for the review. All of them were retrospective. A total of 2498 patients were included in the studies, of whom 734 were treated by mini-invasive approaches and 1764 patients were treated by open surgery. Conclusions: Surgery is the treatment of choice for ACC. Open adrenalectomy (OA) is defined as the gold standard. In recent years laparoscopic adrenalectomy (LA) has gained more popularity. No significant differences were reported for overall recurrence rate, time to recurrence, and cancer-specific mortality between LA and OA, in particular for Stage I-II. Robotic adrenalectomy (RA) has several advantages compared to LA, but there is still a lack of specific documentation on RA use in ACC.

## 1. Introduction

Adrenocortical carcinoma (ACC) is a rare malignancy with an estimated incidence of 1–2 cases per million persons annually [1,2,3,4,5]. It is more frequent in women than in men, with an average age at presentation in the fifth to seventh decades [6,7,8]. The majority of ACC are sporadic cases, often discovered as incidentalomas, whereas less than 10% have found to be associated with genetic syndromes such as Li-Fraumeni, Lynch syndrome, multiple endocrine neoplasia (MEN) type 1, and familial adenomatous polyposis [9,10,11]. In a recent Japanese nation-wide survey, ACCs accounted for 1.4% of a large number of incidentalomas [12]. Due to possible, but not constant, adrenocortical hormone production (mainly cortisol), the clinical presentation of ACC is extremely variable, and most patients (80%) [13] are asymptomatic at the time of diagnosis. Conversely, symptomatic ACC may be diagnosed during clinical evaluation of hypercortisolism, hyperandrogenism, or due to “mass effect” (compression of surrounding structures, pain and/or bleeding [14,15]. An extensive clinical and radiological evaluation of the adrenal mass is clearly the first step to plan the best treatment in case of suspicions of malignancy (Figure 1). Malignancy must be suspected when the tumor size ranges from 4 to 6 cm, with an increased risk for malignancy for masses larger than 4 cm. In particular, an adrenal mass smaller than 4 cm can be considered at low risk of malignancy (around 2%), while a tumor size between 4 and 6 cm has a medium risk of malignancy (6%), and for lesions more than 6 cm in diameter the risk increases to 25% [16]. Moreover, an adrenal nodule increasing in size of more than 1 cm per year must be considered as potentially malignant. In the presence of a suspected ACC, a comprehensive hormone evaluation is mandatory before surgery, in order to clarify the functional diagnosis and preoperative medical treatment. Typically, malignant adrenal functional masses produce inactive steroid precursors or a combined hypersecretion of cortisol and androgens, or several steroidogenic enzymes that could result in increased cortisol production [17]. Imaging features of ACC consist of large irregular shape lesions, internal necrosis or hemorrhages, calcifications, spread/infiltration to adjacent organs and regional lymph nodes and vascular invasion [18,19]. Computed tomography (CT) has been reported to have an optimal diagnostic accuracy to recognize benign or malignant adrenal masses with a sensitivity and specificity of more than 90% [20]. Typical CT features of ACC are poorly defined margins, with heterogeneous contrast enhancement and areas of necrosis or calcifications potentially due to intralesional hemorrhages [21]. Magnetic resonance imaging (MRI) may characterize ACCs as well with high sensitivity (85–100%) and specificity (92–100%) [22]. ACC are reported as isointense or hypointense in T1-weighted sequences, while a high signal intensity is reported for T2-weighted sequences [23]. Areas of hemorrhage may result in variable signal intensity; heterogeneous enhancement can be identified during administration of gadolinium with contrast enhancement and slow wash-out [17]. Recently, 18-FDG PET/CT has been reported as a precise imaging modality for ACC staging with a sensitivity, specificity, and accuracy of about 100% for first diagnosis, and of 98.4%, 100%, and 99.5%, respectively, for re-staging [24]. Although lesion detection by PET/CT and CT was similar, PET/CT may be preferred for localization of potential metastases and follow-up [25]. Previously, the role of fine needle biopsy for the diagnosis of suspected ACC has been considered controversial, due to its low sensitivity and specificity [26], and due to the potential risk of tumor capsule rupture and seeding of tumor cells along the needle route, as well as iatrogenic complications [27]; nowadays, the most recent guidelines, for the same reasons, state that adrenal biopsy is contraindicated and it might be indicated only in oncologic patients to exclude or prove an adrenal metastasis [4]. At time of first diagnosis, ACC is often at an advanced stage, with distant metastases found in 20% of patients, with prevalent location in the lungs and liver (45% and 40%, respectively) [28,29,30]. Thus, ACC prognosis is poor, with an overall survival (OS) of 3.21 years from diagnosis [31]. Five-years survival rates of 82%, 58%, 55%, 18% have been reported for stages I, II, III, IV, respectively [22]. Different results were reported by a multi-registry large case study on 2014 ACC cases, where the median overall survival was 17 months (these poor results are probably justified by the carcinomas’ high grades at diagnosis). The European Network for the Study of Adrenal Tumors (ENSAT) classification is widely accepted to define the ACC stage (Table 1) [32,33]. Surgery remains the treatment of choice for Stage I-III ACC, whereas for Stage IV ACC, surgery may be indicated for palliation [4]. Open surgery has been recognized as the gold standard for ACC because of better achievement of R0 resection. Nevertheless, minimally invasive surgery (MIS), which includes laparoscopic adrenalectomy (LA) and robotic adrenalectomy (RA), is widely accepted as the preferred approach for the resection of benign adrenal tumors, although in the recent years its use in malignant adrenal tumors’ management has been strongly debated.

The aim of this review is thus to analyze the indications for surgery in patients affected by ACC, provide a complete overview on surgical approaches discussing on pros and cons for the open, laparoscopic and robotic techniques. Since the oncological efficacy of MIS for ACC is still in question, this article critically evaluates the optimal surgical approach depending on ENSAT stage and verifies the criteria for correct oncological resection.

## 2. Methods

This comprehensive review has been carried out according to the methodological criteria reported in the Preferred Reporting Items for Systematic Reviews and Meta-Analysis (PRISMA) statement [35] (Figure 2) and was registered on Review Registry Database (reviewregistry1151). The literature sources were the databases PubMed, Scopus and Cochrane Library. The research was focused on the following issue: which kind of surgery should be proposed for which patients affected by suspected/confirmed adrenocortical carcinoma, depending on stage, size and preoperative features? Which are the expected results of open, laparoscopic and/or robotic approaches? The search thread was: ((surgery) OR (adrenalectomy)) AND (adrenocortical carcinoma) Research was limited to 2010, adult patients and papers written in English, with exclusion of review papers. The literature search was performed independently by three authors (AP, DC and SS). Any discrepancies between the reviewers were discussed and solved by consensus. Quality assessment of retrieved studies was performed with JADAD scores in the case of randomized clinical trials [36], or MINORS scores for non-randomized studies [37].

## 3. Results

The review was carried out as a systematic review and focused on open and mini-invasive techniques for the treatment of ACC and limited to 2010 (Table 2). Among all studies identified (Figure 2—PRISMA), 17 were selected for the review. All of them were retrospective. A total of 2498 patients were included in the studies, of whom 734 were treated by mini-invasive approaches and 1764 patients were treated by open surgery. In three studies [38,39,40], MIS included both a laparoscopic and a robotic approach. Among all included studies, ACC staging was performed basing on ENSAT stage: four studies involved stage I–II ACC, seven and six studies involved patients with stage I–III and stage I–IV disease, respectively. Patients treated with open adrenalectomy (OA) were more likely to have a larger tumor: median size was from 6.8 to 14 cm for ACC treated by OA, and from 5.5 to 9 cm for tumors treated by mini-invasive surgery. Seven studies showed that LA was effective for ACC when the tumor size is <10 cm and showed no local invasion, enlarged lymph-nodes or distant metastases (ENSAT stage I–II) [39,40,41,42,43,44,45]. Some authors suggested that LA should be only performed in high volume referral centers. Twelve studies reported a total of 89 conversions from MIS to an open approach [38,39,40,41,42,43,44,45,46,47,48,49]. Conversely, eight further studies [38,46,47,49,50,51,52,53] confirmed OA as the standard operative technique for ACC. Cooper [50] and Leboulleux [53] claimed that LA for ACC is associated with a high rate of recurrence, particularly for peritoneal carcinomatosis. Sixteen studies showed a R0 resection rate from 52 to 100% for ACC treated by OA, and from 50 to 100% for tumors treated by mini-invasive surgery. Five studies compared MIS vs. open lymph node dissection (LND) and one study demonstrated the superiority of traditional surgery when compared to MIS (*p* = 0.01) [39]. Vanbrugghe et al. [48] recommended open multi-organ resection for larger tumors (>12 cm) and in cases of invasive ACC. Several studies reported detailed follow-ups. The rate of local recurrence ranged from 0 to 72% for ACC treated by OA, and from 4 to 55.5% for tumors treated by mini-invasive surgery. The overall recurrence rate ranged from 22 to 69% for ACC treated by OA, and from 13 to 85.7 for tumors treated by mini-invasive surgery. The median disease-free survival (DFS) ranged from 8.1 to 48 months for ACC treated by OA, and from 9.7 to 72 months for tumors treated by mini-invasive surgery. Detailed information is summarized in Table 3.

### 3.1. Laparoscopic Adrenalectomy vs. Open Surgery

In the early 2000s LA was believed to be related with a higher rate of recurrence [53]. Cooper et al. in their retrospective study analyzed 302 patients, 46 of whom underwent laparoscopic adrenalectomy and 256 open surgery. The rate of positive margin was higher and the peritoneal recurrence-free survival shorter in the LA group (*p* = 0.006) [50]. In a retrospective review, Miller et al. showed that 50% of patients who underwent LA had positive margins or intraoperative tumor spillage versus 18% of those who underwent open adrenalectomy. Local recurrence was 25% versus 20%, for LA and OA, respectively [51]. Recent retrospective studies conclude that LA is safe and feasible for tumors <10 cm without evidence of local invasion [40,43]. Porpiglia et al. [54] analyzed 43 ACC patients with I or II ENSAT stage dividing patients into two groups: LA versus OA. No significant differences were reported in terms of recurrence rates and recurrence-free survival among the two groups (50% for LA vs. 64% for OA, *p* = 0.3; 18 months vs. 23 months, *p* = 0.8, respectively). Limited data evaluating the oncological efficacy of LA versus OA for the surgical management of ACC are present in literature. Studies over the last 20 years comparing minimally invasive surgical approaches to open adrenalectomy are reported in Table 2, while surgical and oncological outcomes are described in Table 3. All the published studies are retrospective or non-randomized. In detail, no significant differences were observed in terms of overall survival (OS) and disease-free survival (DFS) between the open and laparoscopic approaches for treatment of ACC, as reported in Table 3. In favour of this consideration, a recent metanalysis by Autorino et al. [55] reported no significant differences for overall recurrence rate (*p* = 0.53), time to recurrence (*p* = 0.11), and cancer-specific mortality (*p* = 0.08) between LA and OA, whereas only peritoneal carcinomatosis was significantly associated with LA (RR 2.39, *p* < 0.001), concluding that OA should be considered the standard surgical management of ACC. Despite the fact ACC treated by LA were significantly smaller in size than the those treated by OA (weighted mean difference, WMD = −3.41 cm, *p* < 0.001) with a higher proportion of Stage I–II for LA (80.8%) compared to OA (67.7%), no differences were reported for operative times (*p* = 0.85) as well as for post-operative complications (*p* = 0.14), whereas significant differences emerged for hospitalization time in favor of LA (WMD = −2.5 days, *p* < 0.001). Thus, an appropriate surgical resection is a mandatory step in the therapeutic management of ACC [4] and the role of minimally invasive surgery is still under investigation [56]. On the contrary, Mpiali et al. in a systematic review, analyzed 1171 patients included between 1999 and 2017 [57]. LA resulted equivalent to OA in terms of R0 resection rate, overall recurrence, disease-free survival and overall survival, however, no data about long time outcomes were presented. OA is considered the treatment of choice for ACC also follow the metanalysis as presented by Xu and colleagues who compared OA vs. minimally invasive surgery (MIS) [58]. Although no significant differences were found for OS (HR 0.97, *p* = 0.801), cancer-specific survival (HR 1.4, *p* = 0.869) and recurrence/disease free survival (HR 0.96, *p* = 0.791) between the two approaches, MIS was significantly associated with earlier recurrence (WMD −8.42, *p* = 0.048), positive surgical margin (RR 1.56, *p* = 0.018) and peritoneal recurrence (RR 2.63, *p* < 0.001). 

### 3.2. The Potential Role of Robotic Approach

The role of robotic surgery in ACC is still under debate. RA has been shown to have several theoretical advantages when compared to LA. To the best of our knowledge, no specific or dedicated studies about RA performed for ACC have been published yet. Data regarding adrenalectomy performed for ACC with minimally invasive robotic techniques are extracted from more general studies. The small number of patients treated with RA for ACC are heterogeneous and does not allow any statistical analysis, so the results must be considered in light of the present limitations. Agcaoglu et al. performed 62 adrenalectomy for tumors larger than 5 cm (24 robotic vs. 38 laparoscopic) showing significant shorter operative time (159.4 ± 13.4 vs 187.2 ± 8.3 min, *p* = 0.043), less conversion rate (4% vs. 11%, *p* = 0.43) and shorter hospital stay (1.4 ± 0.2 vs. 1.9 ± 0.1 days, *p* = 0.009), respectively, concluding that in large masses (>6 cm), RA allowed one to shorten operative time providing less conversion rate compared to LA [59]. Also Nordenstorm et al., in a series of 100 robotic assisted laparoscopies, showed a conversion rate of 7%, but all converted cases were during the initial stage of the robotic approach [60]. This consideration is also in line with a recent meta-analysis about RA and LA [61]. For the primary endpoint, open conversion analysis, results showed significant lower rate for robotic approaches in 18 studies and 1809 patients in total (*p* = 0.02). Although RA and LA showed similar operating times (*p* = 0.18), hospital stays were significantly lower for the RA group (WMD: 0.52; *p* = 0.001). No significant differences on oncological efficacy (*p* = 0.81) and morbidity profile (*p* = 0.94) were reported, and perioperative mortality rate was similar among the groups (*p* = 0.45). The above-mentioned pooled analysis showed a superiority of RA regarding conversion rate and hospital stay compared to LA, but comparable results are provided for operating time, positive margin rate, and postoperative morbidity and mortality [61].

### 3.3. Extension of Surgical Resection and Lymphnode Dissection

Regardless of the surgical approach used, there is a general agreement about the rules of oncologic surgery: “R0 resection en bloc”, “complete excision”, “no tumor grasping or fragmentation or tumor capsule effraction” [16,22,61]. The importance of R0 resection is emphasized by a recent study of 165 patients from 13 United States centers who underwent adrenalectomy for ACC. R0 resection was achieved in 76.4% of patients and surgical margin status was an independent predictor of overall survival. The 5-year overall survival for R0 versus R1 resection was 64.8% and 33.8% (*p* < 0.001) and the 5-year recurrence-free survival for R0 and R1 resection was 30,3% and 13,8% (*p* = 0.03), respectively [62]. This concept is supported by numerous former studies starting back in 1999, when Schulick and Brennan reported the outcomes of 113 patients who underwent to surgical resection for ACC. Sixty eight patients with a complete primary resection had a median survival of 74 months while 45 patients with incomplete primary resection had a median survival of only 12 months (*p* < 0.001) [63].

Last ESMO-EURACAN guidelines advise that locoregional lymphadenectomy improved tumor staging leading to a better oncological outcome [4]. Moreover the ESE guidelines suggest performing a locoregional lymphadenectomy in cases of highly suspected or proven ACC [3]. In the first study regarding LND in ACC, the German ACC Registry analyzed 283 patients: 47 cases underwent adrenalectomy with LND and 236 patients underwent adrenalectomy with no LND. Multivariate analysis indicated a reduced risk of tumor recurrence (hazard ratio [HR] 50.65, P5.42) and disease-related death (HR 50.54, P5.049) for the LND group [64]. More recently, Gerry et al. divided 120 patients into two groups: 32 (27%) received LND and 88 (73%) did not receive LND. Factors related to LND were tumor size (12 cm versus 10 cm, *p* = 0.07), lymph node involvement detected by preoperative imaging (44% versus 7%, *p* < 0.001) and multivisceral resection (78% versus 36%, *p* < 0.001). Multivariate analysis showed that overall survival at five years improved in patients receiving LND [65,66]. In a recent study, Deschner et al. demonstrated that LND is not associated with an increased survival rate. Lymph node metastasis is associated with advanced tumors (*p* = 0.4). Median overall survival was incrementally worse with increasing number of positive lymph nodes (88.2 months for N0, 34.9 months for 1–3 positive nodes, and 15.6 months for ≥4 positive nodes, *p* < 0.001) [67]. The optimal extent of lymphadenectomy in ACC is still not known.

## 4. Discussion

Although surgery remains the treatment of choice for ACC, the role of minimally invasive approaches is still debated in terms of oncological outcomes. In early 2000, the First International Adrenal Cancer Symposium defined open adrenalectomy (OA) as the gold standard for ACC [68]. According to these recommendations, OA represents the treatment of choice to secure oncological principles, as complete R0 “en bloc” resection and lymphadenectomy [69], as also confirmed by the last guidelines [4]. However, since laparoscopic adrenalectomy (LA) was first introduced in 1992 by Gagner [70], its indications have rapidly expanded, including even the treatment for ACC in very selected cases. Even so, the main concern in LA is the risk of capsule rupture and intraperitoneal tumor spread [71]. Current guidelines from ESMO-EURACAN [4] suggest performing LA in patients in unilateral adrenal masses with radiological findings suspicious of malignancy and a diameter ≤ 6 cm, but with no evidence of local invasion (ENSAT stage I/II). Due to the lack of literature concerning the approach for ENSAT stage III, OA still remains recommended for unilateral adrenal masses with radiological findings suspicious of malignancy including signs of local invasion [72]. Furthermore, there is no consensus on the role of LA for tumors >6 cm and local invasion. Gaujoux et al. [30] stated that LA can be considered in experienced centers for tumors with diameter of 5–8 cm without invasion of adjacent organs. Center volume and surgical experience play a crucial role in the outcome of patients with ACC; the last guidelines stated that adrenal cancer surgery should be performed only in centers performing at least six adrenalectomies per year (but with a preference for >20 surgeries per year) [4] and by surgeons with expertise in both open and laparoscopic surgery [73,74].

More recently, robotic adrenalectomy (RA) has also been considered in the management of ACC. Unfortunately, no precise studies about RA performed for ACC are published yet and only few articles report and compare RA to LA or OA in the management of ACC. The first RA was reported in 2008 by Zafar et al., for a 8 cm adrenocortical carcinoma [75]. Since then, the management of adrenal surgery has rapidly evolved and RA has become at least a reliable alternative to LA; however, scientific evidence for the contributions of RA to concrete benefits are still debated. Different technical approaches are available such as robotic-assisted lateral transabdominal adrenalectomy and robotic-assisted posterior retroperitoneoscopic adrenalectomy [76,77]. The transperitoneal approach is very advisable for the larger working space, the easier orientation, and its magnification of surrounding anatomical structures. The retroperitoneal approach mimics OA and should be preferred in the case of bilateral tumors or previous abdominal surgeries.

Beyond the choice of surgical approach, resection should be extended, in the case of extra-adrenal invasion, in order to include en bloc resection of macroscopically invaded surrounding organs: liver, IVC, kidney, pancreas, spleen, stomach and colon [57,78]. In the early 1990s, Icard et al. [79] advocated for en bloc removal of the ipsilateral kidney including peri-hilar lymph nodes and other adjacent structures, to obtain wide operative margins with a low risk of surgical tumor infringement. Thirteen patients (32%) underwent extensive resections, over a 12-year period, including one partial pancreatectomy, four nephrectomies, three right hepatectomies and three bowel resections, all for apparent invasion. Additionally, 11 patients underwent en bloc nephrectomy without clear tumor invasion. No improvement in the outcome was observed. However, the authors argued that an “en bloc resection” allowed for R0 surgery. An Italian review also failed to find a survival advantage for concurrent nephrectomy associated to adrenalectomy for ACC without evidence of direct tumor invasion [80]. Kidney involvement is rare and there is no evidence that nephrectomy may positively influence the oncologic outcome. It is suggested to remove an adjacent organ on a case-by-case basis, considering preoperative imaging and overall inspection during surgery. Tumor thrombi may not be considered as a contraindication to resection [81]. Main vein tumor thrombi can potentially be present in most patients with T4 tumors. In a retrospective review, Laan et al. [82] compared survival in patients undergoing resection with (*n* = 28) or without (*n* = 37) inferior vena cava tumor thrombi. The authors found similar rates of complete resection, short-term postoperative morbidity and overall survival at 12 and 24 months. However, in the subgroup of study with only complete resection, survival rates became disparate, ranging from 36 through 60 months (40% vs. 0% in patients with and without resected tumor thrombi, respectively). Another important issue is the use of a no-touch technique and ensuring no tumor fragmentation during en bloc resection. Some studies have shown very high recurrence rates and poor overall survival after tumour rupture or spillage [34,83]. Fragmentation is easy to induce because ACC is a soft and friable tumor. Some authors state that direct contact tumor is possible only with an open approach [8,16]. There is no consensus on the role of lymph-node dissection (LND) in adrenal tumors. The adrenal gland has two main lymphatic drain flows: the first to the inferior vena cava and right/left edge of aorta and the second one to the lomboaortic nodes and interaorticocaval space. Adrenal lymphatic drainage patterns are complex, so the extention of lymphadenectomy in ACC resection remains unclear [84]. Standardization of regional lymphadenectomy has been proposed by Gaujoux et al. to include first-level drain nodes; nevertheless, it has not been widely adopted [69]. In the absence of clear-cut evidence of any benefits in terms of oncologic outcome, extended resection should be performed in selected cases, when lymph node involvement is detected on preoperative imaging or intraoperatively.

The limitations of this study and the difficulty to draw conclusions from the evaluated studies are due to multiple confounding factors. All the studies analyzed were retrospective and included few cases due to the overall rarity of ACC. Furthermore, despite the fact this comprehensive review included studies from 2010, lots of these papers analyzed patients operated on over a period long before 2010, when radiological imaging for staging was perhaps less accurate and surgical approaches were certainly different, with minor numbers of minimally invasive operations. Patients enrolled were in different stages; many patients underwent MIS were in the initial stage (I–II), while advanced stages (III–IV) were often treated by the open approach. LND and complete resection conferred better oncologic outcomes, but they are not standardized and depend on the stage of presentation and surgeons’ expertise. On the other hand, lots of patients included in the previous studies were operated at low-volume centers. It would have been useful to compare surgical managements grouping tumors by size, hormonal profile or other clinical characteristics, but these data are poorly available and inhomogeneous among the studies. Lastly, to the best of our knowledge, no studies about RA performed for only ACC are published yet. Thus, clear indications and unambiguous management of ACC patients is still lacking. Further investigations, with patient randomization according to staging and surgical treatment, are therefore needed.

## 5. Conclusions

The suspicions of ACC for an adrenal lesion are driven by tumor size (>6 cm), radiological signs of malignancy of ^18^FDG-PET uptake, presence of local invasion or distant metastases and typical hormonal secretions (i.e., androgen, oestrogen, and steroid precursors)**.** Surgery is the treatment of choice for ACC (Stage I–III) whereas for Stage IV ACC surgery may be of more palliative intent (Figure 3). During the last years surgical approaches have changed. Initially OA has defined as the gold standard for confirmed or suspicious ACC. LA has gained more consensus for its indications and efficacy. No significant differences were reported for overall recurrence rate, time to recurrence, and cancer-specific mortality between LA and OA, in particular in Stage I–II cases. Theoretically, robotic adrenalectomy has been showed to have several advantages when compared to LA, but there is still a lack of documentation of RA on malignant adrenal lesions, thus no direct conclusion about RA in ACC can be inferred. The importance of R0 resection is emphasized by several studies, with en bloc removal of adjacent involved tissues or organ for locally advanced lesions. Current guidelines state that locoregional lymphadenectomy improved tumor staging and a better oncological outcome can be reached, while there is no consensus about the extent of lymphadenectomy. An appropriate surgical resection is a mandatory step in the therapeutic management of ACC and although RA represents the future prospective, the role of minimally invasive surgery still needs further investigation.

## Figures and Tables

**Figure 1 biomedicines-09-00909-f001:**
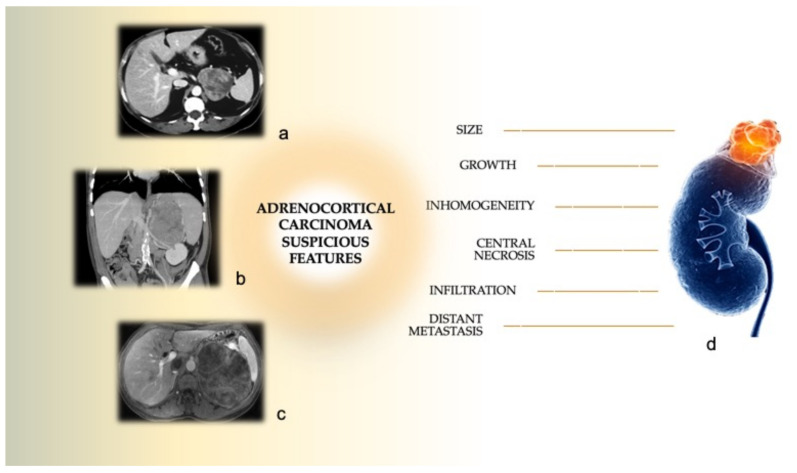
Imaging features suspicious for adrenocortical carcinoma. (**a**) CT scan, axial view, showing a left adrenal 7 cm inhomogeneous mass. (**b**) CT scan, coronal view of a huge mass infiltrating the spleen. (**c**) MRI showing a left 18 cm lipid-rich ACC. (**d**) Suspicious features for adrenocortical carcinoma (Adapted from dreamstime.com).

**Figure 2 biomedicines-09-00909-f002:**
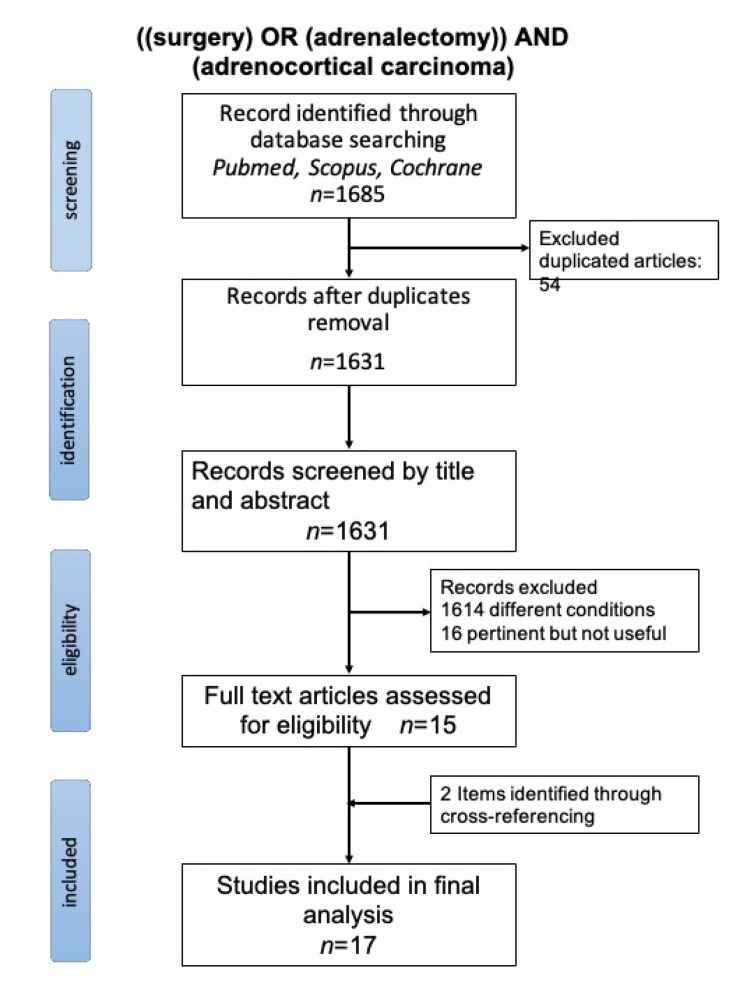
Prisma flowchart.

**Figure 3 biomedicines-09-00909-f003:**
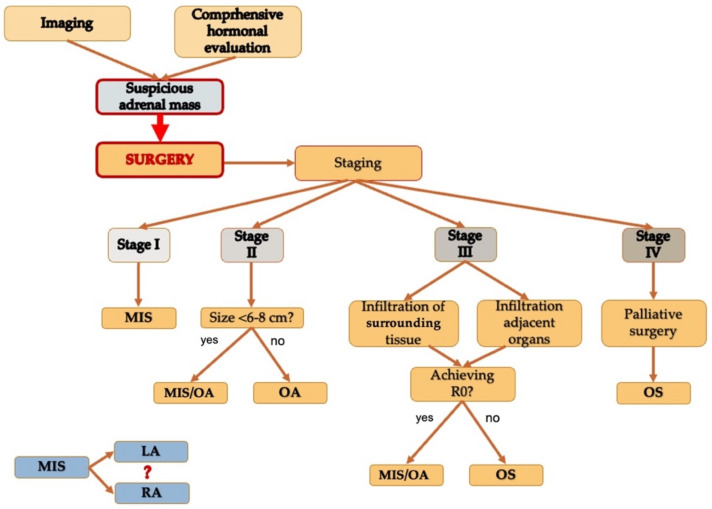
Conclusions: Surgical treatment and management of adrenocortical carcinoma. MIS, Minimally Invasive Surgery; OA, Open Adrenalectomy; OS, Open Surgery; LA, Laparoscopic Adrenalectomy; RA, Robotic Adrenalectomy.

**Table 1 biomedicines-09-00909-t001:** ENSAT Stages for Adrenocortical Carcinoma [34].

ENSAT Stage	TNM	Description
**I**	T1 N0 M0	The tumor has 5 cm or less of major diameter (T1), it has not spread to nearby lymph nodes (N0) or distant sites (M0).
**II**	T2 N0 M0	The tumor is greater than 5 cm and it has no grown into tissues outside the adrenal gland (T2), it has not spread to nearby lymph nodes (N0) or distant sites (M0).
**III**	T1 N1 M0	The tumor has 5 cm or less of major diameter and it has not grown into tissues outside the adrenal gland (T1). The cancer has spread to nearby lymph nodes (N1) but not to distant sites (M0).
	T2 N1 M0	The tumor is greater than 5 cm and it has not grown into tissues outside the adrenal gland (T2). The cancer has spread to nearby lymph nodes (N1) but not to distant sites (M0).
	T3 Any N M0	The tumor is growing in the fat surrounding the adrenal gland. The tumor can be any size (T3). It might or might not have spread to nearby lymph nodes (Any N0). It has not spread to distant sites (M0).
	T4 Any N M0	The tumor is growing into nearby organs (kidney, pancreas, spleen and liver or large blood vessels such as renal vein or vena cava). The tumor can be any size (T4). It might or might not have spread to nearby lymph nodes (Any N). It has not spread to distant organs (M0).
**IV**	Any T Any N M1	The cancer has spread to distant sites like the liver or lungs (M1). It can be any size (Any T) and may or may not have spread to nearby tissues (Any T) or lymph nodes (Any N).

**Table 2 biomedicines-09-00909-t002:** Characteristics of the studies.

Study	Study Period	Study Design	Minors Score	Patients *n*	Median Age (Year)	Gender F M *n*, (%)	Hormon Secretion *n*, (%) OA, LA	Surgical Approach *n*, (%)	Conversion *n* (%)	Median Follow-Up Months, OA: MIS
Kastelan et al., (2020) [41]	2004–2018	Retrospective	18	46	48 (18–74)	32 (69) 14 (31)	11:10 (48:43)	0A 23 (50) LA 23 (50)	0	52
Zheng et al., (2018) [46]	2013–2015	Retrospective	19	42	46 (40–54)	23 (55)	13:11	0A 22 (52) LA 20 (48)	0	Maximum 36
Wu et al., (2018) [47]	2009–2017	Retrospective	20	44	45 (2–74)	27 (61) 17 (39)	9:11 (39:52)	0A 23 (52) LA 21 (48)	1	34
Calcatera et al., (2018) [38]	2010–2014	Retrospective	17	588	54	360 (61) 228 (39)		0A 388, (66) MIS 200, (34)	38 (19)	
Maurice et al., 2017 [39]	2010–2013	Retrospective	17	481	OA 56 (43–67) LA 61 (50–69)	302 (63) 179 (37)		OA 320, (67) MIS 161, (33)	24 (15)	25:23.6
Lee et al., 2017 [40]	1994–2014	Retrospective	17	201	52 (11–87)	131 (65) 70 (35)	58:11 40:25	0A 154, (77) MIS 47, (23)	9 (19)	60
Vanbrugghe et al., 2016 [48]	2002–2013	Retrospective	18	25	47 (22–77)	15 (60) 10 (40)	3:4 (33:25)	0A 9, (36) LA 16, (64)	0	52.9:36.4
Donatini et al., 2014 [43]	1985–2011	Retrospective	21	34	45	26 (76) 8 (24)	8:3 (38:23)	0A 21, (61)L A 13, (39)	0	66
Mir et al., 2013 [49]	1993–2011	Retrospective	18	44	49 (40–65)	22 (50) 22 (50)		0A 26, (59) LA 18, (41)	5 (24)	26
Fossa et al., 2013 [44]	1998–2011	Retrospective	17	32	48 (29–75)	23 (72) 9 (28)	6:13	0A 15, (47) LA 17, (53)	2 (11)	29.1
Cooper et al., 2013 [50]	1993–2012	Retrospective	17	302	45	196 (65) 106 (35)		0A 256, (85) LA 46, (15)	-	34.4
Miller et al., 2012 [52]	2005–2011	Retrospective	18	156	47 (10–80)	64 (41) 92 (59)		OA 110, (70) LA 46, (30)	-	29.5:19
Lombardi et al., 2012 [45]	2003–2010	Retrospective	21	156	47 (10–81)	100 (64) 56 (36)	62	0A 126, (80) LA 30, (20)	0	42
Porpiglia et al., 2010 [54]	2002–2008	Retrospective	19	43	43 (24–68)	26 (60) 17 (40)	14:11 (56:61)	0A 25, (58) LA 18, (42)	-	35
Miller et al., 2010 [51]	2003–2008	Retrospective	16	88	46 (18–81)	57 (65) 31 (35)		0A 71, (81) LA 17, (19)	-	36.5
Leboulleux et al., 2010 [53]	2003–2009	Retrospective	17	64	54 (23–79)	36 (56) 28 (44)	35 (30)	0A 58, (90) LA 6, (10)	-	35
Brix et al., 2010 [42]	1996–2009	Retrospective	19	152	51	10844	63:34	0A 117, (77) LA 35, (23)	11 (34)	39.3

OA, open adrenalectomy; MIS, minimally invasive surgery (Robotic and Laparoscopic adrenalectomy); LA laparoscopic adrenalectomy.

**Table 3 biomedicines-09-00909-t003:** Surgical and oncological outcomes.

Study	Tumor Stage (ENSAT)	Tumor Size (OA: MIS) cm, Median	R0 Resection (OA: MIS) *n*, (%)	LND (OA: MIS) *n*	Local Recurrence (OA: MIS) *n*, (%)	Overall Recurrence (OA: MIS) *n*, (%)	Disease Free Survival (OA: MIS) Median, Months, (%)	Overall Survival (OA: MIS) Months (%)
Kastelan et al., (2020) [41]	I–III	12:7.5	23:23 (100:100)	-	2:1 (9:4)	5:3 (22:13)	- *p*:0.55	- *p*:0.76
Zheng et al., (2018) [46]	I–III	10.1:6.3	22:20 (100:100)	-	5:8 (23:40)	13:11 *p*:0.08	45:17 *p*:0.02	-
Wu et al., (2018) [47]	I–II	6.8:5.8	-	3:0	5:9 (22:43)	12:11 (52:52)	22:25 (36:39) *p*:0.8	42:63 (43:47) *p*:0.63
Calcatera et al., (2018) [38]	I–IV	12.4:8.9	289:141 (74:70)	-	-	-	-	-
Maurice et al., 2017 [39]	I–IV	11.7:7.5	266:129 (83:80)	**42:2** ***p*:0.01**	-	-	-	(62:58) *p*:0.42
Lee et al., 2017 [40]	I–IV	10.9:5.5	114:36 (74:77)	63	-	82:22 (64:48) *p*:0.07	10:14 (3.8:9.1) *p*:0.2	53.8:90.9 (49:68) *p*:0.23
Vanbrugghe et al., 2016 [48]	I–III	11.6:6.2	9:12 (100:75)	-	0:2 (0:12)	4:6 (44:37)	(62:56) *p*:1.0	(89:69) *p*:0.36
Donatini et al., 2014 [43]	I–II	6.8:5.5	21:13 (100:100)	-	-	5:4 (24:31)	47:46	(81:85) *p*:0.63
Mir et al., 2013 [49]	I–IV	13:7	16:11 (61:61)	14:6	12:10 (46:55)	(27:22)	13.8:9.7 (60:39)	(54:58) *p*:0.6
Fossa et al., 2013 [44]	I–III	13:8	12:12 (80:70)	-	1:1 (7:6)	5:3 (33:17)	8.1:15.2	36:103 *p*.0.22
Cooper et al., 2013 [50]	I–IV	12:8	134:25 (52:71)	-	-	73:76.1	16.7:10.9	110:54 *p*:0.07
Miller et al., 2012 [52]	I–III	12:7.4	72:26 (65:56)	-	-	(40:86)	-	**Stage II 103/51 *p*:0.002** Stage III 44/28 *p*:0.77
Lombardi et al., 2012 [45]	I–II	9:7.7	126:30 (100:100)	23:1	14:4 (11:13)	48:8 (38:26)	48:72 (38:58)	- (47:66) *p*:0.2
Porpiglia et al., 2010 [54]	I–II	10.5:9	25:18 (100:100)	-	6:6 (24:33)	16:9 (64:50)	18:23	(72:95)
Miller et al., 2010 [51]	I–III	12.3:7	(82:50)	-	(20:25)	(65:63)	19:10	-
Leboulleux et al., 2010 [53]	I–IV	14:7	37:5 (63:83)	-	(72:34)	-	20	38:5 -
Brix et al., 2010 [42]	I–III	8:6.2	64:24 (55:69)	-	(38:50)	81:27 (69:77)	21.5–24.2	-

OA, open adrenalectomy; MIS, minimally invasive surgery (Laparoscopic and Robotic adrenalectomy); LND, lymph node dissection; DFS, disease-free survival; OS, overall survival; Significant differences are represented in bold.

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
