# Peer review of "Surgical Management of Adrenocortical Carcinoma: Current Highlights"

_biomedicines, 2021, doi:10.3390/biomedicines9080909_

Round 1

Reviewer 1 Report

The manuscript entitled “Surgical management of adrenocortical carcinoma: current highlights” by Cavallaro et al present a comprehensive review with a focus on surgery methods (open adrenalectomy [OA], and minimally invasive surgery [MIS] which includes [laproscopic adrenalectomy [LA] and robotic adrenalectomy [RA]) for the treatment of adrenocortical cancer (ACC). The study was carried out according to the guidelines of PRISMA. Seventeen retrospective studies were selected from Pubmed, Scorpus and Cochrane Library for the study. The authors discuss that OA is the gold standard method with LA gaining more indications. The authors report that there were no significant differences in overall recurrence rate, time to recurrence, and cancer-specific mortality between the two procedure methods, particularly for stages I and II. They also report that RA has several advantages than LA, and warrants additional studies to confirm this. This systematic review is useful with an impact on clinical practice. However, there are some major and minor recommendations to improve.

Major Comments:

  1. The authors have not provided the cortisol and androgen levels in the studies mentioned. If available, they must provide the range of these hormone levels in all the studies mentioned and correlate with different grades of ACC.
  2. Authors must provide information on the number of females and males in each study, the type of mutation – is it sporadic incidentalomas or associated with genetic syndromes? Provide correlations between the sex, type of mutations and prognosis and recovery.

Minor Comments:

  1. Throughout the study, authors have used several abbreviations without providing explanation at the first mention. For e.g., authors have provided the abbreviation for open adrenalectomy as OA in the discussion while they have used the term OA several times before that. Authors must explain briefly the terms used.
  2. All figure legends must be explained in more detail. Figure 1 doesn’t have any explanation except a title for the figure. In figure 3, explain ‘si’ and ‘no’.
  3. What is T1-weighted sequence and T2-weighted sequence of MRI? Explain in a sentence or two.
  4. Explain R0 resection and R0 ‘en bloc’
  5. There are typos in lines 50, 94 and in table 1.

Author Response

Dear Editor, Dear Reviewers,

We would like to sincerely thank you for the detailed review of our manuscript, your valuable suggestions, and your precise comments.

Please find responses to your comments and consecutive changes based on your recommendations below.

Thank you again.

Reviewer 2 Report

The manuscript of Cavallaro et al is a comprehensive review of the surgical management of adrenocortical carcinoma. There is definite need for more understanding, evidence of what procedure is preferred in which stage(s).

Unfortunately, this review does not bring us much more close to this understanding.

I have major concerns and questions to be answered.

1/ This review fails to refer to the last, and most recent review regarding the adrenocortical carcinomas: ESMO guideline 2020, Fassnacht et al Annals of Oncology. The review of ENSAT / ESE 2018, Fassnacht et al EJE 2018 is cited, but not on the surgical issues, more on other findings, such as the incidence of ACC (which is much better to be retrieved from a study directly studying the incidence by Kerkhofs et al Eur J Cancer 2013 49 (11) 2579. In short, these guidelines should be cited in the introduction, and the conclusions of this review should be compared to these guidelines. Does this review add to the knowledge from these guidelines, which are not always with the greatest evidence.

2/ The discussion is loaded with results and difficult to tread and understand. Please bring these results to that section, and perhaps refer to studies in the tables. The discussion than can be upon the question how to operate stage I and II, and stage III. What is the evidence, and what are the expert opinions combined with observations from the studies. Etc.

I would be interested to show clearer the results of RA and compared to LA. Als a separate paragraph on it in the discussion.

3/ What is missing from the literature, and this review is not to blame of course, are direct comparisons between OA and LA and/or RA. This review would bring us further if they would be able to better compare cases and characteristics of the tumors operated on. So is it possible to compare tumors in the cited papers with same sizes, other characteristics and perhaps same hormonal profile. If this is possible it would be a little step forward. If this is not possible (probably the case) the authors could discuss this.

4/ The review takes studies from the last 20 years into account. The patients will be, in some of them, from longer before. Are these historical cases to be compared in this review? Was for example the radiology well enough to stage the patients before operation well enough? Please explain the readers if this is the case or not, and perhaps make a comment in the limitations.

Minor concerns

5/ Throughout the manuscript the abbreviations OA, LA and RA are used, but not explained when used first.

6/ In line 77 fine needle biopsy is considered controversial. That is not the case. It is contraindicated except for diagnosing metastases. Refer to the 2 guidelines.

7/ The citation of prognosis of ACC is somewhat old. Is there a better and more recent paper to refer to?

8/ LA should be performed in high volume referral centers is mentioned in the results section and the discussion. Could you put more arguments and citation to this statement? See also the guidelines. NB also for OA it is mandatory in my opinion that the surgery takes place in experience hands by surgeon’s working in a multidisciplinary team. (Also guidelines).

9/ I find table 3 difficult to understand

10/ In the introduction and discussion it is stated that ACC is the most common primary malignancy of the adrenal gland. Is that true? Pheochromocytomas have a higher incidence. The percentage of malignancy is difficult to define, as only metastatic disease is considered as proof of malignancy. Other tumors, such as lymphoma’s I would not count them as primary tumor of the adrenal gland.

Author Response

(The authors gave the same response as above.)

Round 2

Reviewer 2 Report

Thank you for the revised manuscript. I will respond per point made.

Ad 1/ sufficient

Ad 2/ I do not understand, and disagree on the answer. Results in the relevant section.

Ad 3/ OK

Ad 4/ studies are from 2010 and have patients of much older dates. To be corrected throughout the text. Limitation is OK.

Ad 5/ OK

Ad 6/ fine needle biopsy changes OK

Ad 7/ OK

Ad 8/ OK

Ad 9/ improved

Ad 10/ one of the most common malignancy of the adrenal gland. Delete “one of the most”. It is a nonsense remark most or one of the most. It is just ACC.

Additionally:

  1. Figure 1c, could it be a myelolipoma, so no ACC?
  2. Figure 3. Decision to do MIS is made obligatory if R0 can be achieved and size <6-8cm. Not correct. Not according to current guidelines
  3. Studies were not from the last 10 years (2010 is longer ago).
  4. RA has no documentation on malignant adrenal lesions. So do. Not mention that more studies comparing LA and RA are needed for ACC. (Conclusions!)

Author Response

Thank you for the revised manuscript. I will respond per point made.

REPLY: Dear Reviewer we want to extend to you our most sincere gratitude. We have replied point by point.

Ad 1/ sufficient

REPLY 1: Thank you.

Ad 2/ I do not understand, and disagree on the answer. Results in the relevant section.

REPLY 2: Thank you and sorry for the misunderstanding. We have moved the results (and p values) in the proper section, dividing the section in sub-chapters dealing with OA versus LA, potential role of RA and Role of Lymph Node Dissection.

Ad 3/ OK

REPLY 3: Thank you.

Ad 4/ studies are from 2010 and have patients of much older dates. To be corrected throughout the text. Limitation is OK.

REPLY 4: Thank you. We have modified the text accordingly.

Ad 5/ OK

REPLY 5: Thank you.

Ad 6/ fine needle biopsy changes OK

REPLY 6: Thank you.

Ad 7/ OK

REPLY 7: Thank you.

Ad 8/ OK

REPLY 8: Thank you.

Ad 9/ improved

REPLY 9: Thank you.

Ad 10/ one of the most common malignancy of the adrenal gland. Delete “one of the most”. It is a nonsense remark most or one of the most. It is just ACC.

REPLY 10: Thank you for the comment. We are sorry for the mistake. We have delete the sentence.

Additionally:

  1. Figure 1c, could it be a myelolipoma, so no ACC?

REPLY 11: Thank you for your comment. This was one of our most interesting cases of an ACC lipid rich, as confirmed by histology. We have modified the figure caption accordingly.

  1. Figure 3. Decision to do MIS is made obligatory if R0 can be achieved and size <6-8cm. Not correct. Not according to current guidelines

REPLY 12: Thank you again for your insights. Current guidelines explain that MIS is advisable whereas in the past OA was mandatory. Figure 3 was modified.

  1. Studies were not from the last 10 years (2010 is longer ago).

REPLY 13: Thank you. We have modified the text accordingly.

  1. RA has no documentation on malignant adrenal lesions. So do. Not mention that more studies comparing LA and RA are needed for ACC. (Conclusions!)

REPLY 14: Thank you again for your input and we have modified the conclusion accordingly.

Round 3

Reviewer 2 Report

The re-revised version is much better regarding the presentation of the results and the consequences for the discussion.

There remain 3 points which should be corrected or carified before I would advise to accept the paper.

1/ Does the reviewed literature go to 20 years? Please be correct in this, check.

2/The section on RA does not state that there are no reports, or perhaps report of one patient with that technique. So it is all theoretical for ACC. The results for RA were all, or perhaps with very few exclusions on benign adrenal tumors. The text in that part should be adapted, the conclusion should be made in accordance.

3/ Lymfnode resection is advised (ESMO), guideline, and suggested (ESE guideline) with the uncertainty to which extend. Acknowledging that the evidence for better overall survival is not clear, but the better loco-regional control is more so. The result section on lymfnodes should be adapted, and the statement in the conclusion also.

NB the above mentioned adaptations should be reflected in the abstract accordingly.

Author Response

REPLY: Dear Reviewer, we would like to thank you for your overall positive evaluation.

1/ Does the reviewed literature go to 20 years? Please be correct in this, check.

REPLY 1: / The review considered studies from 2010, and in some of them, patients are included from longer before. We hope we have clarified this aspect. The manuscript has been revised accordingly.

2/The section on RA does not state that there are no reports, or perhaps report of one patient with that technique. So it is all theoretical for ACC. The results for RA were all, or perhaps with very few exclusions on benign adrenal tumors. The text in that part should be adapted, the conclusion should be made in accordance.

REPLY 2: Dear Reviewer, we agreed with you and with your insight. We have modified the draft in particular RA paragraphs and conclusion, according to your suggestion.

3/ Lymfnode resection is advised (ESMO), guideline, and suggested (ESE guideline) with the uncertainty to which extend. Acknowledging that the evidence for better overall survival is not clear, but the better loco-regional control is more so. The result section on lymfnodes should be adapted, and the statement in the conclusion also.

NB the above mentioned adaptations should be reflected in the abstract accordingly.

REPLY 3: Dear Reviewer, thanks again for your insights. We have improved the LN section including this consideration in the manuscript.

Round 4

Reviewer 2 Report

The content of the corrections is now good for me. The spelling/English of the new content should be corrected.